# Nutrient Intake from Voluntary Fortified Foods and Dietary Supplements in Japanese Consumers: A Cross-Sectional Online Survey

**DOI:** 10.3390/nu15143093

**Published:** 2023-07-10

**Authors:** Chiharu Nishijima, Yoko Sato, Tsuyoshi Chiba

**Affiliations:** 1Department of Preventive Gerontology, Center for Gerontology and Social Science, National Center for Geriatrics and Gerontology, Obu City 474-8511, Japan; 2Department of the Science of Living, Kyoritsu Women’s Junior College, Tokyo 101-8437, Japan; yoksato@kyoritsu-wu.ac.jp; 3National Institute of Health and Nutrition, National Institutes of Biomedical Innovation, Health, and Nutrition, Osaka 566-0002, Japan; tyschiba@nibiohn.go.jp

**Keywords:** vitamins, minerals, voluntary fortified foods, dietary supplements, nutrient intake

## Abstract

Consuming voluntary fortified foods (vFFs) and dietary supplements (DSs) is one method for addressing micronutrient inadequacy, but their efficacy is unclear. This study explored the use of vFFs and DSs, and the role of package labels. We conducted a nationwide online survey of 4933 Japanese adults in 2020. The use of vFFs and DSs was 23.3%. The reported vFFs were cereal products (*n* = 370), milk products/milk substitutes (*n* = 229), and one-shot drinks (*n* = 144). Vitamins, calcium, and iron were the main micronutrients added to the vFFs. Most users consumed vitamins C and B from vFFs and/or DSs, and females also consumed iron. The median intake of vitamins B6 and C, selenium, and zinc (plus vitamin B2 and copper in females) exceeded 50% of the respective estimated average requirement values. Approximately 60–70% of the users referred to nutrition labels. However, only a small percent of the respondents clearly understood the nutrient content. To address insufficient nutrient intake, the use of vFFs and DSs may be a good alternative; however, consumer education on the use of vFFs/DSs and package labels needs to be implemented before encouraging their use.

## 1. Introduction

Micronutrients, such as vitamins and minerals, are important determinants of human health. Nutrient-poor diets can result in poor health outcomes. Food fortification with vitamins and minerals is one of the strategies for addressing micronutrient malnutrition suggested by the World Health Organization and the Food and Agriculture Organization of the United Nations [1]. The mandatory fortification of commonly consumed foods has been practiced in 143 countries [2].

In Japan, mandatory food fortification has not been implemented. Instead, many voluntary fortified foods (vFFs) are marketed in various forms, such as dairy products, soft drinks, snack foods, and staple cereal products. People can consume these foods as replacements for unfortified conventional foods. Although recent studies on Japanese adults have reported that significant proportions of the population do not meet the recommended levels of intake of some vitamins and minerals, the amounts of vitamins and minerals added to vFF products were not considered in many cases [3,4,5]. Because fortified foods are not listed on the Japanese food composition table, detailed product information must be included in the dietary record to calculate added vitamins and minerals.

In the limited number of studies that reported vFF use in Japan, the prevalence of vFF consumers is reported to be as low as a few percent using the dietary record method [6]; however, approximately 30% was reported by asking about usage habits, regardless of whether the survey was conducted in person, by questionnaire, or online [7,8,9]. The reasons for these differences are not clear, but the dietary record used in the former study was a one-day survey. The previous studies in other countries have used the food frequency questionnaire, and assessing habits may be more appropriate to clarify vFF use [10,11]. In this regard, the prevalence and usage of vFFs in Japan are unclear.

Studies on the nutrient intake from vFFs in Europe report that vFFs make a significant contribution to mean daily intake of Irish adults and the Dietary Reference Intakes (DRIs) of micronutrients in Polish adolescents [11,12], and a study in the United States has reported the risks of excessive intake [13]. Thus, the prevalence of inadequacy observed in Japanese adults may be overestimated, and the prevalence of intakes above the tolerable upper intake level (UL) may be underestimated [3,4,5].

Furthermore, in addition to dietary supplements (DSs) for vitamin and mineral supplementation that have been traditionally marketed, DSs for other purposes, including those containing protein and amino acids, fatty acids, flavonoids, and herbs, are also fortified with vitamins/minerals. However, it is not known whether consumers are aware of the added vitamins/minerals in DSs for purposes other than that of vitamin/mineral supplementation. Currently, the vitamin and mineral intake from vFFs and DSs is not well understood.

Despite the limited studies on vFFs and DSs [7,8,9,11,12,13], we hypothesized that vFFs and DSs substantially contribute to the dietary intake of vitamins and minerals. Therefore, the present study explored the use of vFFs and DSs among Japanese adults and estimated nutrient intake from these additional sources. In this study, we defined DSs as any food in the form of tablets, capsules, or powders that contain added vitamins/minerals (i.e., both vitamin/mineral supplements (VMSs) and DSs for other purposes) [8,9,11]. We also examined the use of package labels by vFF and DS consumers for purchase selection and understanding the nutrient content.

## 2. Materials and Methods

### 2.1. Online Survey Procedure and Study Population

This study was performed in accordance with the Declaration of Helsinki and approved by the Ethical Review Board of the National Institutes of Biomedical Innovation, Health, and Nutrition (Ikikenhatsu-32020; approval date: 7 January 2020). The nationwide, cross-sectional survey was conducted by Intage Inc. (Tokyo, Japan), an online research company that manages 4.3 million authenticated potential respondents nationwide. The company manages its potential respondents by sharing personal information with a cell phone carrier provider or sending a postcard to the physical address as an identity verification process.

The survey was conducted between 7 February and 12 February 2020. The survey was divided into two phases: a screening survey to identify vFF and DS users and a full-scale survey to collect detailed information. An invitation email for survey cooperation with a webpage link to the survey form was sent to 34,939 computer-randomized survey monitors aged 20 to 79 years. The population was divided into 12 cells by sex and 10-year age groups, and at least 400 responses from each cell were collected. An explanation of the study was provided on the first page of the survey form, and only those individuals who agreed to participate answered the questionnaire. In the screening survey, the responses were collected on a first-come, first-served basis until the number reached 400 for each 10-year age group for males and females. There was a total of 4933 participants in the screening survey (collection rate: 14.1%). This collection rate was similar to another Japanese nationwide online survey [14]. Information about sex, age, residential area, and marital status was obtained from the survey company using the registered data.

### 2.2. Questionnaire

The questionnaire consisted of 3 questions in the screening survey and 10 questions in the full-scale survey. The estimated time to complete the questionnaire was around 10 min. In the screening survey, we asked, “Do you eat/drink the vitamins or minerals added products such as the products claim to contain vitamins or minerals on the package?” with five illustrations of foods with nutrient content claims. This question was developed based on the preceding survey to ascertain the words in which consumers most correctly identify vFF products. The response options were “Almost every day”, “3 to 4 times a week”, “1 to 2 times a week”, “2 to 3 times a month”, “less than once a month”, “never eat/drink”, and “not sure”. Therefore, we considered people who responded with a frequency of more than 2 to 3 days a month as the users; “less than once a month” and “never eat/drink” as the non-users, and “not sure” as the ones who were not sure. The dietary type was determined based on the participant’s characteristics and included weight loss, vegetarian/vegan, low salt/sodium, weight gain, high protein, low fat, high carbohydrate, low carbohydrate, and no special diet [15]. Subjective health was asked because it is reported that dietary supplement use was associated with health status [16]. The subjective health status of response options was “Good”, “Have some health complaints”, and “Have a chronic disease”.

In the full-scale survey, we asked about the names, manufacturers, and targeted nutrients of the vFF and DS products that the participants most often used (up to three products) as product information [17]. Among the 4933 respondents of the screening survey, 948 males and 1214 females self-identified as users of vFFs and/or DSs, but the analysis of product information found that 839 of them reported conventional foods (not nutrient-fortified foods) or drug/quasi-drug products; thus, they were classified as part of the non-users group. In addition, an analysis of product information found that 172 self-identified users did not remember the product information; thus, they were classified as the not sure group to compare the characteristics of those who are aware of the use of vFFs and/or DSs and those who are not. Finally, the vFF and/or DS users in this study comprised 461 males (18.7%) and 690 females (28.0%).

For usage information and the use of the package labels, we asked “Number of products in use”, “Frequency of use”, “Duration of use”, “Reasons for use”, “Factors that affect purchase selection”, “Label items that influence purchase selection”, “Understanding of nutrient contents”, and “Calculation of the total amount of nutrients” in a choice question form with response options that shown in the tables. 

### 2.3. Product Information Analysis

The products were identified by Google searching for product information, and label information was obtained from the manufacturers’ websites [17]. This process was conducted by trained dietitians, and the results were confirmed by C.N. When products were found to have several flavors with similar vitamin or mineral contents, the most common flavor was used. The amount of nutrients consumed through vFFs and DSs per day was estimated by multiplying the frequency and nutrient content labeled on the packages. We identified the added nutrients using the ingredient list, but the labeled amounts of nutrient content for vFFs did not differentiate between the added and naturally occurring amount, and the nutrient content reflected the total amount. The median value was used for calculation when the nutrient content was declared to be within a range. For the quantity of products consumed, we did not ask how much participants consumed, and existing data were not sufficient to estimate the general intake of most food items; therefore, we assumed that participants consumed products in the suggested amount written on the label or, for beverages, one bottle. For comparison, the values of the estimated average requirement (EAR) and tolerable upper intake level (UL) of DRIs for Japanese (DRIs-J) 2020 were used as references [18].

### 2.4. Statistical Analyses

Participants were first classified into user, non-user, and not sure groups, and the distributions of characteristics were expressed as percentages; the differences between groups were compared using the chi-squared test according to sex. Regarding the user group, the amounts of vitamins and minerals from vFFs and DSs were presented as the 50th, 75th, 90th, and 97.5th percentiles. Since no significant differences by age were observed in the preliminary analysis, we differentiated the participants only by sex. The usages of vFFs and DSs and the use of the package labels among the user group were compared between “vFF-only”, “DS-only”, and “vFF + DS” subgroups. Differences were examined using the chi-squared test. Statistical analyses were performed using JMP (SAS Institute Inc., Cary, NC, USA) version 13.0 with the statistical significance set to *p* < 0.05.

## 3. Results

### 3.1. Characteristics

Table 1 shows the characteristics of 4933 participants according to the use of vFFs and/or DSs in a full-scale survey. More of the younger participants were users of vFFs and/or DSs in both males and females. Of all participants, 41.0% of males and 32.1% of females were not sure about the use of vFFs and DSs. Marital status was significantly different only in males. More participants in the not sure group were relatively older and had no special diet.

### 3.2. Nutrient Contents of vFF and DS Products per Daily Suggested Amount

The cumulative number of products reported was 907 vFF and 599 DS products containing added nutrients (Table 2). The most reported vFF was cereal products (370 products); next was milk products/milk substitutes, which include milk, cheese, whey proteins, soy milk, almond milk, and any probiotic yogurt (229 products); one-shot drinks, which are concentrated drinks including energy drinks and jelly drinks (144 products); and vegetable/fruit juice (72 products). In terms of the ingredient information, all vitamins, calcium, and iron were added to the vFFs. Most (>60%) cereal products contained added vitamins A, B1, B2, B12, niacin, folic acid, and iron; most milk/milk products contained added calcium; most one-shot drinks and vegetable/fruit juices contained vitamin C; and most of the other beverages contained vitamins B6 and C, with a variation in nutrient content of 4 to 200 times the recommended amount.

Regarding DSs, participants reported 470 individual or multiple vitamin/mineral supplements (VMSs) and 129 other supplements, such as those for weight loss, beauty, and eye health purposes (Table 3). The nutrient contents on the label in the reported DS products varied depending on the products, but all vitamins were contained in almost half of VMS products. Minerals such as calcium, magnesium, iron, selenium, copper, and zinc were labeled in approximately 20% of the VMS products. 

### 3.3. Distribution of Nutrient Intakes from vFFs and DSs

#### 3.3.1. Amounts Compared to DRIs-J

The distribution of nutrient intake from vFFs and/or DSs according to sex is shown in Table 4. More than half of the participants consumed vitamins B1, B2, C, niacin, and folic acid from vFFs and/or DSs in both males and females, and also consumed vitamin B12 and iron in females. The median intake of vitamins B6 and C, selenium, and zinc (plus vitamin B2 and copper in females) exceeded 50% of the respective EAR values. The remaining vitamins and most of the remaining minerals (magnesium, iron, molybdenum, copper, and iodine) accounted for 20% to 40% of the respective EAR values. The 90th percentile in females and 97.5th percentile in males for vitamin A, the maximum intake of zinc in males, and vitamin B6, niacin, folic acid/folate, iron, and zinc in females exceeded the respective UL values.

#### 3.3.2. Amounts According to the Use of vFFs and/or DSs

The distribution of nutrient intake from vFFs and/or DSs according to the use of vFFs and/or DSs the participants consumed is shown in Table 5. Among all users, 699 (60.7%) were vFF-only users, 388 (33.7%) were DS-only users, and 64 (5.6%) were both vFF and DS (vFF + DS) users. Over half of the DS-only and vFF + DS users consumed all the vitamins, while half of the vFF-only users consumed more specific vitamins, such as vitamin B1, C, niacin, and folic acid/folate. Among minerals, calcium and iron were consumed by 35.8% and 62.2% of the vFF-only users, respectively. Both minerals were consumed by approximately 30% of the DS-only users, and over 60% of the vFF + DS users. The amounts of the median intake of nutrients in DS-only users were 3 to 10 times higher than those in vFF-only users for most nutrients. When comparing the maximum intakes, the amounts were the highest for all the listed nutrients in DS-only users. The amounts of vitamins A and B6, niacin, folic acid, iron, and zinc consumed by DS-only users exceeded the ULs of DRIs-J.

### 3.4. Usages of vFFs and/or DSs

Most participants reported that they used one to three vFF and/or DS products (Table 6). The frequency of vFF product usage by vFF-only users varied from almost every day to two to three times per month, while over half of the DS-only and vFF + DS users used products almost every day (*p* < 0.001). One-third of vFF-only users reported their use for less than six months, whereas approximately 60% of DS-only and vFF + DS users reported their use for over three years (*p* < 0.001). The most and second-most common reasons for use among all groups were “I feel like I am undernourished” and “I feel that my nutritional balance is bad.” However, “I want to eat what I like without worries about a balanced diet” and “I want to choose a nutritious food” were the third and fourth reasons for vFF-only and vFF + DS users while “uncomfortable health condition appeared” was the third reason for DS-only users. The reasons for use were significantly different in most response options.

### 3.5. Use of the Package Labels for Purchase Selection and Understanding of Nutrient Contents

A higher content of nutrients was reported as the most desirable factor for purchase selection in DS-only and vFF + DS users (*p* = 0.015), while the favorable taste was the most important for vFF-only users (*p* < 0.001) (Table 7). Lower prices were similarly important regardless of the use of vFFs and/or DSs; however, foods with health claims and popular or much-talked-about items had more influence on purchase selection in vFF + DS users (*p* = 0.022 and *p* = 0.012, respectively). Eye-catching words on the front of the package and expiration dates were selected more by vFF-only users (*p* < 0.001 and *p* = 0.006, respectively). Nutrition labels, product name, ingredients, and functional claims were selected more by vFF + DS users (*p* = 0.002, *p* = 0.015, *p* = 0.038, and *p* < 0.001, respectively). Warning labels were selected more by both DS-only and vFF + DS users (*p* < 0.001). Although 60–70% of individuals, regardless of the use of vFFs and/or DSs, referred to nutrition labels, only 10% of DS-only and vFF + DS users and 4% of vFF users clearly understood the nutrient content. Approximately 80% of users never calculated the total amount of nutrients they consumed, regardless of the use of vFFs and/or DSs.

## 4. Discussion

In this study, we explored the use of vFFs and DSs among Japanese adults. We hypothesized that vFFs and DSs contribute substantially to the dietary intake of vitamins and minerals. Nutrient intake from vFFs and/or DSs fulfilled more than half of the EAR for some vitamins/minerals for most users. However, a few users exceeded the ULs. Our study suggests the importance of assessing nutrient intake, including the use of vFFs and DSs.

This study found that 23.3% of participants used vFFs and/or DSs. This prevalence was higher than what was previously reported, but similar to that in a more recent report of Japanese adults [6,8]. However, this prevalence was quite low compared with that of other populations, such as adults in the United States [13], Ireland [19], Finland [20], and adolescents in Poland [11]. The predominant vFF products consumed by the participants in this study were cereal and dairy products. Among cereal products, breakfast cereals accounted for the most, although breakfast cereals are not as commonly consumed in Japan as in Western countries [21]. The consumption of milk and dairy products was more common (70% of the Japanese population) than that of breakfast cereals but still less common than that in the United States and Ireland [4,22,23]. Therefore, differences in dietary habits may have partly influenced the low prevalence of vFF use in our study. It should also be noted that one-third of our participants were not sure if they used vFFs and/or DSs. Food products with added vitamins/minerals are constantly being developed, thus there may be unconscious users [24]. Further studies are required to quantify the unconscious use.

Currently, in mandatory food fortification programs, cereal grain (maize, rice, or wheat), a staple food source, is the most commonly fortified food [2]. In Japan, rice, bread, and noodles are the most consumed staple foods [21], and some are voluntarily fortified with vitamins such as folic acid and are available in most supermarkets. However, this was not reported in the present survey. Under the circumstances of unmandated fortification, the adoption of vFFs in dietary habits depends on consumers. To consume more vitamins/minerals than those in conventional foods, it may be easier to use vFFs as an additional food source, such as snacks and drinks, rather than as meals. However, it is well known that the consumption of snacks and sugar-sweetened drinks contributes significantly to daily energy intake. In addition, small meals with snacks are common among women in some countries [25]. Since it has been suggested that the consumption of nutrient-poor snacks may be associated with high body mass index, eating in the absence of hunger, eating away from home or work, social modeling, and food insecurity [25], the association of vFFs (nutrient-rich snacks) with these outcomes should be studied.

According to previous research, fortification with iron, magnesium, zinc, vitamin A, and B vitamins helped to reduce the prevalence of inadequacy in the UK population [26]. Research on a Japanese population found that the daily intake of vitamins A, B1, and calcium in more than half of males and females did not meet EAR [4]. Although it cannot be compared with the values obtained from different participants with different methodologies, their median intake of vitamins A, B1, and calcium could be increased above the EAR if the median values of the respective nutrients that our users took from vFFs and/or DSs were added. These results suggest that the use of vFFs and/or DSs could reduce the prevalence of inadequate nutrient intake.

Because DSs contain more nutrients and are easier to consume regularly than vFFs, excess intake of some nutrients has been reported [10,12,13]. In our study, a few users with intakes exceeding the ULs of vitamin A, B6, and iron among the DS + vFF and DS-only users were observed. Therefore, the use of vFFs is only safer when there is caution regarding excess intake. In addition, for vitamins A and B6, in which excess intakes were observed, the median intake of vFFs may be enough to prevent inadequate daily nutrient intake compared with that of the daily intake of Japanese representatives [4]. For iron, the use of vFFs only (median intake of 1.1 mg/day) may not be sufficient for menstruating women (10% contribution to EAR); however, the use of DSs may only increase the risk of excessive intake. Thus, it is important to be cautious of the amounts consumed from vFFs and/or DSs. Although two-thirds of vFF and/or DS users use multiple products, less than 20% calculate the total amount of nutrients every time or occasionally. Since the proportion of those who refer to nutrition labels for purchase selection was already higher among vFF + DS or DS-only users than previously observed [27,28], as a next step, education on considering the total amounts of added nutrients being consumed through vFFs and/or DSs may be needed.

The strength of this study is that this is the first report that focused on the use of vFFs among Japanese adults to clarify the vitamin/mineral intake from vFFs and DSs and their usage. However, the results of this study should be interpreted in light of several limitations. First, owing to the nature of the online survey, our results may be subject to self-reporting and recall biases. In addition, the participants were online survey monitors, and socio-demographic data was not collected; these make generalization difficult. Although the questions about the use and usage of vFFs and DSs were developed while referring the previous studies, they have not been validated. Second, the finding that a considerable number of the participants were not sure about the use of vFFs and/or DSs indicates the presence of unconscious users; this implies a possible underestimation of the prevalence of vFF and/or DS users. Moreover, the question of the screening survey was selected based on the preceding survey; however, the food products that 38.8% of self-reported vFFs and/or DSs users were not nutrient-fortified foods. This is partly because consumers usually choose these foods by reading claims (e.g., high in vitamin C), which cannot tell whether the claimed vitamins and minerals are fortified or naturally occurred until they read the ingredient information. In this regard, we assumed that self-report overestimates the vFFs use. The use of other methods such as the food purchase diary [20] may be considered to obtain more accurate information. Third, the vFF and/or DS users in this study may be biased toward individuals with a more health-oriented lifestyle, as health consciousness is a common characteristic of vFF and/or DS users [19,20,24,29]. Fourth, the study did not include a question about the daily amount of consumption. The questionnaire should have included the question “how much do you consume?” as compared to the daily serving size indicated on the package; there may have been participants who consumed less or more vFF and/or DS products than the daily serving size indicated on the packages. This implies bias in the estimation of nutrient intake. In addition, this study focused only on vFF and/or DS consumption without considering regular food, which did not allow the estimation of the contribution of nutrient intake from these products as a proportion of the total intake. This implies that the real figures will be higher (with the possible associated health risks), and that the possible continuity of the study requires an assessment of the complete dietary intake.

## 5. Conclusions

In this study, 23.3% of participants were vFF and/or DS users, with nutrient intakes fulfilling more than half of the EAR for some vitamins/minerals for most users. To address insufficient nutrient intakes, the use of vFFs and DSs may be a good alternative, but consumer education on the use of vFFs and/or DSs and their package labels needs to be implemented before encouraging vFF and DS use. To reduce the prevalence of inadequate nutrient intake, the use of vFFs and preferably DSs for iron for certain people may be suggested with attention to the contribution of vFFs to daily energy intake and the total amount of nutrients from DSs. Future studies to investigate the contribution of vFFs and DSs in the total intake of nutrients are warranted.

## Figures and Tables

**Table 1 nutrients-15-03093-t001:** Characteristics of participants in a full-scale survey according to the use of vFFs and/or DSs ^1^.

	Males			*p*-Value ^2^	Females			*p*-Value
	Users (*n* = 461)	Non-Users (*n* = 993)	Not Sure (*n* = 1012)		Users (*n* = 690)	Non-Users (*n* = 984)	Not Sure (*n* = 793)	
Proportion of groups	18.7	40.3	41.0		28.0	39.9	32.1	
Age				<0.001				<0.001
20 to 39	41.4	31.7	30.4		41.0	28.0	32.3	
40 to 59	35.8	31.5	34.4		33.8	33.7	32.8	
60 to 79	22.8	36.8	35.2		25.2	38.3	34.9	
Marital Status				<0.001				0.381
Not married	47.7	36.8	39.9		36.1	31.9	36.4	
Married	52.3	63.2	60.1		63.9	68.1	63.6	
Residential Area ^3^				0.418				0.625
Urban	59.0	56.4	56.1		59.0	57.5	56.2	
Rural	41.0	43.6	43.6		41.0	42.5	43.8	
Dietary Type				<0.001				<0.001
Weight loss	4.8	3.8	4.6		4.2	4.0	3.2	
Vegetarian/vegan	2.2	2.4	1.2		1.6	1.6	2.9	
Low salt/sodium	5.6	8.4	5.4		7.4	8.1	4.4	
Weight gain	23.4	21.5	18.2		10.1	11.7	11.6	
High protein	12.1	9.6	7.2		10.0	10.3	5.6	
Low fat	3.5	1.9	1.4		2.5	3.2	2.7	
High carbohydrate	8.7	5.7	4.7		15.9	11.1	9.8	
Low carbohydrate	2.6	2.9	2.8		2.8	2.7	1.4	
No special diet	37.1	43.8	54.6		45.5	47.4	58.5	
Health status				0.051				0.138
Good	58.8	60.5	62.4		58.4	61.2	61.3	
Have some health complaints	33.8	29.8	27.0		34.4	30.4	32.9	
Have a chronic disease	7.4	9.7	10.7		7.3	8.4	5.8	

^1^ Expressed in percentage. ^2^ Differences in distribution between groups were examined by chi-squared test. ^3^ Urban areas comprise prefectures that have ≥5 million inhabitants. Abbreviations: vFF, voluntary fortified food; DS, dietary supplement.

**Table 2 nutrients-15-03093-t002:** Nutrient contents of vFF products per daily suggested amount.

	Cereal Products ^1^				Milk/Milk Products ^2^				One-Shot Drinks ^3^				Vegetable/Fruit Juice				Other Food Products ^4^				Other Beverages ^5^			
	*n* = 370 ^6^				*n* = 229				*n* = 144				*n* = 72				*n* = 29				*n* = 63			
	*n* ^7^	% ^8^	Min	Max	*n*	%	Min	Max	*n*	%	Min	Max	*n*	%	Min	Max	*n*	%	Min	Max	*n*	%	Min	Max
Vitamin A ^9^	275	74.3	96	661	5	2.2	33	550	22	15.3	82.5	1069	13	18.1	495	1225	4	13.8	383	600	0	–		
Vitamin B1	369	99.7	0.2	0.8	25	10.9	0.1	1.5	61	42.4	0.1	2.3	16	22.2	0.6	2	2	6.9	1.0	1.2	8	12.7	3.4	5.3
Vitamin B2	247	66.8	0.1	1.0	25	10.9	0.1	1.0	67	46.5	0.1	5.0	7	9.7	1.4	1.8	4	13.8	1.1	1.1	9	14.3	0.3	9.5
Niacin	364	98.4	2.1	11	22	9.6	0.6	15	70	48.6	1.5	30	6	8.3	13	13	2	6.9	11	13	35	55.6	4.6	50
Vitamin B6	140	37.8	0.2	1.0	30	13.1	0.1	2.7	75	52.1	0.1	5.0	16	22.2	0.4	1.3	3	10.3	1.0	1.1	47	74.6	0.4	8.6
Vitamin B12	236	63.8	0.2	1.9	74	32.3	0.2	3.9	25	17.4	0.4	12	7	9.7	2.4	6.2	2	6.9	2.0	2.7	2	3.2	0.6	30
Folic acid/folate	259	70.0	31	140	118	51.5	8	350	22	15.3	45	513	11	15.3	93	533	6	20.7	60	270	1	1.6	300	300
Vitamin C	204	55.1	15	50	22	9.6	14	65	129	89.6	32	1500	65	90.3	38	1000	10	34.5	15	500	60	95.2	15	3000
Calcium	94	25.4	7	460	180	78.6	60	700	7	4.9	11	227	15	20.8	40	135	11	37.9	33	450	6	9.5	13	225
Magnesium	15	4.1	41	110	5	2.2	18	140	6	4.2	22	40	14	19.4	45	45	0	–			5	7.9	4.6	22
Iron	363	98.1	1.2	6.8	127	55.5	0.6	10.5	11	7.6	0.8	5	7	9.7	2.7	20	7	24.1	4	22	2	3.2	3.2	6.4
Selenium	0	–			0	–			1	0.7	12	12	0	–			0	–			0	–		
Molybdenum	0	–			0	–			1	0.7	8.1	8.1	0	–			0	–			0	–		
Copper	0	–			0	–			2	1.4	0.1	0.3	0	–			0	–			0	–		
Zinc	0	–			0	–			5	3.5	2.0	2.9	0	–			2	6.9	10	14	0	–		
Iodine	0	–			0	–			1	0.7	1.0	1.0	0	–			0	–			0	–		

^1^ Cereal products include breakfast cereals, cereal bars, and cookies. ^2^ Milk products include probiotic products, cheese, and whey proteins as well as milk substitutes such as soy milk and almond milk. ^3^ One-shot drinks include energy drinks, lemonades, and jelly-type vitamin/mineral drinks. ^4^ Other food products include eggs, fish, sausages, wafers, and jelly. ^5^ Other beverages are mostly marketed as plastic bottle flavored-water beverages such as “C.C. lemon”, “vitamin water”, and “AQUARIUS”. ^6^ Cumulative number of reported products. ^7^ The number of products that contained each of the added nutrients. ^8^ The percentage of products that contained each of the added nutrients among the respective product types. ^9^ The units for each nutrient are as follows; Vitamin A, µg RAE; Vitamin B1, mg; Vitamin B2, mg; Niacin, mg; Vitamin B6, mg; Vitamin B12, µg; Folic acid/folate, µg; Vitamin C, mg; Calcium, mg; Magnesium, mg; Iron, mg; Selenium, µg; Molybdenum, µg; Copper, mg; Zinc, mg; Iodine, mg. Abbreviations: vFF, voluntary fortified food; RAE, retinol activity equivalent.

**Table 3 nutrients-15-03093-t003:** Nutrient contents of DS products per daily suggested amount.

	VMSs				Other Supplements			
	*n* = 470 ^1^				*n* = 129			
	*n* ^2^	% ^3^	Min	Max	*n*	%	Min	Max
Vitamin A (µg RAE)	212	45.1	0.4	4500	15	11.6	1.1	1500
Vitamin B1 (mg)	253	53.8	0.1	100	34	26.4	0.2	18
Vitamin B2 (mg)	315	67.0	0.1	60	37	28.7	0.4	12
Niacin (mg)	252	53.6	1.4	500	10	7.8	3	15
Vitamin B6 (mg)	267	56.8	0.1	100	30	23.3	0.2	20
Vitamin B12 (µg)	271	57.7	0.8	150	22	17.1	0.5	60
Folic acid/folate (µg)	293	62.3	3	1000	11	8.5	40	600
Vitamin C (mg)	299	63.6	0.4	1980	17	13.2	20	666
Calcium (mg)	122	26.0	13	1000	9	7.0	70	400
Magnesium (mg)	99	21.1	8	400	6	4.7	25	200
Iron (mg)	121	25.7	0.2	44	6	4.7	2.5	5.6
Selenium (µg)	105	22.3	2.8	277	6	4.7	4.8	70
Molybdenum (µg)	46	9.8	0.06	80	0	–		
Copper (mg)	95	20.2	0.01	2	3	2.3	0.01	0.54
Zinc (mg)	124	26.4	0.01	30	11	8.5	0.3	13.2
Iodine (mg)	34	7.2	0.2	150	0	–		

^1^ Cumulative number of reported products. ^2^ The number of products that contained each of the added nutrients. ^3^ The percentage of products that contained each of the added nutrients among the respective product types. Abbreviations: DS, dietary supplement; VMS, vitamin/mineral supplement; RAE, retinol activity equivalent.

**Table 4 nutrients-15-03093-t004:** Distribution of nutrient intakes from vFFs and DSs according to sex.

	*n* ^1^	% ^2^	Percentile						
			50	75	90	97.5	Max	EAR ^3^	UL
Male (*n* = 461)									
Vitamin A (µg RAE)	207	44.9	225	770	1772	2936	4050	550–600	2700
Vitamin B1 (mg)	275	59.7	0.3	1.5	2.7	17	100	1.0–1.2	n/e
Vitamin B2 (mg)	262	56.8	0.4	1.7	3.5	13	50	1.1–1.3	n/e
Niacin (mg)	282	61.2	3	11	15	30	100	11–13	300−350
Vitamin B6 (mg)	230	49.9	0.7	2.0	3.2	13	100	1.1	50–60
Vitamin B12 (µg)	223	48.4	0.8	3.0	7.0	40	126	2.0	n/e
Folic acid/folate (µg)	245	53.1	80	200	240	508	800	200	900–1000
Vitamin C (mg)	302	65.5	50	140	323	1000	1980	80–85	n/e
Calcium (mg)	166	36.0	102	207	343	721	1000	600–650	2500
Magnesium (mg)	72	15.6	63	100	125	318	400	270–310	n/e
Iron (mg)	226	49.0	1.9	4.0	5.0	10	21	6.0–6.5	50
Selenium (µg)	63	13.7	23	50	50	74	80	25	400–450
Molybdenum (µg)	24	5.2	8	21	45	80	80	20–25	600
Copper (mg)	61	13.2	0.3	0.6	0.9	2.0	2.0	0.7	7.0
Zinc (mg)	76	16.5	6	12	15	22	45	9	40–45
Iodine (mg)	18	3.9	37	130	150	150	150	95	3000
Female (*n* = 690)									
Vitamin A (µg RAE)	308	44.6	204	450	2700	2700	5400	450–500	2700
Vitamin B1 (mg)	415	60.1	0.3	1.5	4.4	40	47	0.8–0.9	n/e
Vitamin B2 (mg)	369	53.5	0.5	2.2	5.5	30	36	0.9–1.0	n/e
Niacin (mg)	411	59.6	3	11	17	40	500	9–10	250
Vitamin B6 (mg)	317	45.9	0.8	3.2	10.0	30	47	1.0	40–45
Vitamin B12 (µg)	362	52.5	0.8	3.0	10.0	30	126	2.0	n/e
Folic acid/folate (µg)	408	59.1	80	200	240	499	1000	200	900–1000
Vitamin C (mg)	406	58.8	75	189	1000	1291	3000	80–85	n/e
Calcium (mg)	242	35.1	135	340	360	689	1040	500–550	2500
Magnesium (mg)	73	10.6	45	100	156	206	206	220–240	n/e
Iron (mg)	365	52.9	1.9	4.0	6.6	11	44	5.0–11.0	40
Selenium (µg)	48	7.0	15	50	50	243	277	20	350
Molybdenum (µg)	23	3.3	6	17	27	50	50	20	500
Copper (mg)	39	5.7	0.3	0.5	0.8	1.0	1.0	0.6	7
Zinc (mg)	58	8.4	3	11	15	29	30	6–7	30–35
Iodine (mg)	17	2.5	25	58	134	150	150	95	3000

^1^ The number of participants who consumed each added nutrient from vFFs and/or DSs among respective sex. ^2^ The percentage of participants who consumed each added nutrients from vFFs and/or DSs among respective sex. ^3^ Given as a range of values depending on age according to the DRIs for Japanese. Abbreviation; DRIs-J, dietary reference intakes for Japanese; EAR, estimated average requirement; UL, tolerable upper intake level; RAE, retinol activity equivalents; n/e, not established.

**Table 5 nutrients-15-03093-t005:** Distribution of nutrient intake from vFFs and DSs according to the use of vFFs and/or DSs.

	vFF-Only (*n* = 699)							DS-Only (*n* = 388)							vFF + DS (*n* = 64)						
	*n* ^1^	% ^2^	Percentile					*n*	%	Percentile					*n*	%	Percentile				
			50	75	90	97.5	Max			50	75	90	97.5	Max			50	75	90	97.5	Max
Vitamin A ^3^	288	41.2	96	204	257	745	1225	190	49.0	600	1510	2700	3469	5400	37	57.8	378	1445	2861	2957	2957
Vitamin B1	413	59.1	0.2	0.4	0.5	2.0	5.3	234	60.3	1.8	3.8	25.0	40	100	43	67.2	1.3	2.5	7.1	40	40
Vitamin B2	310	44.3	0.1	0.3	1.0	2.4	9.5	277	71.4	2.0	3.5	14.0	32	50	44	68.8	1.8	2.4	7.6	30	30
Niacin	429	61.4	2	4	6	18	50	222	57.2	13	15	40	100	500	42	65.6	7	18	27	40	40
Vitamin B6	269	38.5	0.3	0.7	1.5	4.9	8.6	240	61.9	2.0	3.2	15.0	30	100	38	59.4	1.8	3.2	23	31	31
Vitamin B12	303	43.3	0.4	0.8	1.2	4.2	30	237	61.1	3.0	6.0	20.0	100	126	45	70.3	2.3	6.0	13	78	81
Folic acid/ folate	355	50.8	40	80	120	408	700	248	63.9	200	240	400	652	1000	50	78.1	120	270	320	509	520
Vitamin C	402	57.5	31	112	453	1069	3000	260	67.0	100	300	1000	1050	1980	46	71.9	100	169	650	1433	1450
Calcium	250	35.8	60	238	350	700	1040	117	30.2	200	300	370	801	1000	41	64.1	120	340	401	827	843
Magnesium	38	5.4	10	45	52	90	90	92	23.7	83	123	167	269	400	15	23.4	17	50	116	141	141
Iron	435	62.2	1.1	3.4	5.0	7.9	24	108	27.8	3.4	7.0	10	23	44	48	75.0	3.9	5.0	7.0	12.9	13.5
Selenium	1	0.1	12	12	12	12	12	99	25.5	25	50	50	103	277	11	17.2	6	25	46	50	50
Molybdenum	1	0.1	8.1	8.1	8.1	8.1	8.1	39	10.1	8	25	40	80	80	7	10.9	1	5	25	25	25
Copper	2	0.3	0.02	0.02	0.02	0.02	0.02	88	22.7	0.3	0.6	0.8	1.8	2.0	10	15.6	0.1	0.4	0.9	0.9	0.9
Zinc	6	0.9	1	6	14	14	14	114	29.4	6	13	15	28	45	14	21.9	2	9	14	15	15
Iodine	1	0.1	1	1	1	1	1	30	7.7	39	81	150	150	150	4	6.3	16	104	130	130	130

^1^ The number of participants who consumed each added nutrient from respective vFF and/or DS type. ^2^ The percentage of participants who consumed each added nutrient from respective vFF and/or DS type. ^3^ The units for each nutrient are as follows; Vitamin A, µg RAE; Vitamin B1, mg; Vitamin B2, mg; Niacin, mg; Vitamin B6, mg; Vitamin B12, µg; Folic acid/folate, µg; Vitamin C, mg; Calcium, mg; Magnesium, mg; Iron, mg; Selenium, µg; Molybdenum, µg; Copper, mg; Zinc, mg; Iodine, mg. Abbreviation; vFF, voluntary fortified food; DS, dietary supplement; RAE, retinol activity equivalent.

**Table 6 nutrients-15-03093-t006:** Usage information for vFFs and DSs according to the use of vFFs and/or DSs ^1^.

	All ^2^ (*n* = 1151)	vFF Only (*n* = 699)	DS Only (*n* = 388)	vFF + DS (*n* = 64)	*p*-Value ^3^
Number of products in use					<0.001
1 product	33.9	37.6	32.7	0	
2 to 3 products	58.4	57.8	53.9	92.2	
4 to 5 products	5.5	3.6	9.0	4.7	
≥6 products	2.3	1.0	4.4	3.1	
Frequency of use					<0.001
Almost every day	43.6	31.6	63.1	56.3	
3 to 4 times/week	17.0	19.0	13.9	14.1	
1 to 2 times/week	23.5	28.8	14.2	23.4	
2 to 3 times/month	15.8	20.6	8.8	6.3	
Duration of use					
<6 months	27.2	34.0	17.0	14.1	<0.001
6 months to <3 years	21.9	20.2	25.5	18.8	
≥3 years	50.9	45.8	57.5	67.2	
Reasons for use ^4^					
I feel like I am undernourished.	47.4	42.9	50.5	76.6	<0.001
I feel that my nutritional balance is bad.	38.4	35.6	42.5	43.8	0.054
I want to eat what I like without worries about balanced diet.	26.8	33.6	16.0	18.8	<0.001
I want to choose a nutritious food.	26.0	30.6	17.8	25.0	<0.001
Uncomfortable health condition appeared (e.g., fatigue and rough skin).	18.4	13.0	28.6	15.6	<0.001
It was a product I wanted to eat.	14.6	20.3	4.1	15.6	<0.001
It is a popular product/I heard it is good for health.	7.1	7.2	5.7	15.6	0.016
Other	4.5	3.4	6.4	4.7	0.073

^1^ Expressed in percentage. ^2^ Participants who used at least one or more vFF and/or DS. ^3^ Differences in distribution between groups were examined by chi-squared test. ^4^ Multiple answers were allowed. Abbreviations: vFF, voluntary fortified food; DS, dietary supplement.

**Table 7 nutrients-15-03093-t007:** Use of the package labels for purchase selection and understanding of nutrient contents according to the use of vFFs and/or DSs ^1^.

	All ^2^ (*n* = 1151)	vFF Only (*n* = 699)	DS Only (*n* = 388)	vFF + DS (*n* = 64)	*p*-Value ^3^
Factors that affect purchase selection ^4^					
Higher content of nutrients	54.8	51.8	58.2	67.2	0.015
Favorable taste	48.6	65.8	16.8	53.1	<0.001
Lower price	45.5	43.9	46.4	57.8	0.094
Domestically produced	28.4	28.2	28.1	32.8	0.724
Foods with health claims	24.6	22.6	26.0	37.5	0.022
Additive-free	13.6	12.6	14.4	18.8	0.319
Popular or much-talked-about items	11.8	11.0	11.3	23.4	0.012
Other	2.0	0.6	4.4	3.1	<0.001
Label items that influence purchase selection ^4^					
Nutrition labels	64.7	60.9	69.6	76.6	0.002
Eye-catching words on the front of the package	46.4	51.2	37.9	45.3	<0.001
Product name	37.4	39.9	31.7	43.8	0.015
Expiration dates	31.9	35.2	25.8	32.8	0.006
Ingredients	30.0	27.9	31.7	42.2	0.038
Functional claims	23.4	18.6	28.9	42.2	<0.001
Small letters on the front/back of the package	18.8	17.7	20.1	21.9	0.511
Warning labels	11.0	7.3	16.8	17.2	<0.001
Allergen labels	7.7	8.3	7.2	4.7	0.525
Other	1.5	1.4	1.5	1.6	0.987
Understanding of nutrient contents					<0.001
No	41.8	49.9	27.6	39.1	
Yes, broadly	51.8	46.4	61.6	51.6	
Yes, clearly	6.4	3.7	10.8	9.4	
Calculation of the total amount of nutrients ^5^					0.017
Never	53.3	83.9	74.3	84.4	
Occasionally	11.2	14.2	21.8	15.6	
Every time	1.6	1.8	3.8	0	

^1^ Expressed in percentage. ^2^ Participants who used at least one or more vFF and/or DS. ^3^ Differences in distribution between groups were examined by chi-squared test. ^4^ Multiple answers were allowed. ^5^ The percentage among those who use multiple vFF or DS products. The number of participants in each group: vFF-only, 436; DS-only, 261; vFF + DS, 64. Abbreviations: vFF, voluntary fortified food; DS, dietary supplement.

## Data Availability

The data presented in this study are available on request from the corresponding author.

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
