# Peer review of "Nutrient Intake from Voluntary Fortified Foods and Dietary Supplements in Japanese Consumers: A Cross-Sectional Online Survey"

_nutrients, 2023, doi:10.3390/nu15143093_

Round 1

Reviewer 1 Report

The authors present an interesting topic to evaluate nutrient intake from fortified foods and dietary supplements. They provide information regarding nutrient content of products and motivations for purchasing different fortified or supplement products. The authors provide reasoning for the need of further assessing nutrient intake and consumer education in this population. This is one of few studies in this population that evaluates voluntarily fortified foods and supplements and how this contributes to nutrient intake. However, the paper needs English language revision overall as some portions are unclear and there is missing data. I believe the paper needs revisions before being acceptable for publishing.

Introduction

·         Mention limited related studies – besides the reference in line 40 are there any other related study findings?

·         A small area of improvement would be writing more about meeting dietary recommendations of different nutrients and dietary behavior of Japanese people.

Methods

·         Did the questionnaire contain any questions related to socioeconomic status?

·         Were questions related to use in past years? Or different time frame?

·         Please describe sampling method.

·         Why were not educational status, income, physical activity, smoking, weight, and BIM measured?

·         Why was not dietary intake assessed by dietary questionnaire? Understanding the intake of various nutrients from different food sources is crucial for evaluating the nutritional status of the Japanese population.

·         It is important to assess the energy intake of participants.

Results

·         138: Suggest renaming the section title

·         152-154: Rephrase wording is awkward/unclear

·         Table 3: why only differentiating based on sex?

·         170: rename subheading

·         171 – 182: Table 4 is not shown

·         Table 5: Suggest renaming table “based on type of foods” is unclear

·         Significant p-values not displayed in text when describing results

·         Were there any adverse effects associated with the consumption of fortified foods and, in particular, dietary supplements? Did the participants report any negative outcomes?

Conclusion

·         234-236: This sentence is unclear, separate into two parts

·         285-287: Discussing nutrient adequacy by age, but this was not evaluated in the paper. Only evaluated nutrient adequacy by sex

·         Since the majority of fortified foods, such as beverages, are considered unhealthy, do you advise individuals to increase their consumption of fortified foods? What are your thoughts on dietary supplements? Is it a good strategy to promote the use of dietary supplements?

·         What are the implications of this study?

Minor comment

·         vFF abbreviation – is this for voluntary fortified foods? In abstract different definition used (nutrient-fortified foods) then in introduction (voluntary fortified)

The paper needs English language revision overall as some portions are unclear and there is missing data. 

Author Response

Answer to Reviewer 1

Thank you for valuable comments and useful suggestion to improve our manuscript. We carefully read them and changed our manuscript by following each comment.

Introduction

  • Mention limited related studies – besides the reference in line 40 are there any other related study findings?
  • A small area of improvement would be writing more about meeting dietary recommendations of different nutrients and dietary behavior of Japanese people.

(Response) Thank you for your suggestion. We added more findings from Japan and other countries.

“In the limited number of studies that reported vFF use in Japan, the prevalence of vFF consumers is reported to be as low as a few percent using the dietary record method [4]; however, approximately 30% was reported by asking about usage habits, regardless of whether the survey was conducted in person, by questionnaire, or online [5-7]. The reasons for these differences are not clear, but the dietary record used in the former study was a one-day survey. The previous studies in other countries have often used the food frequency questionnaire, and assessing habits may be more appropriate to clarify vFF use. In this regard, the prevalence and usage of vFFs in Japan are unclear.” (Lines 47-54)

“Studies on the nutrient intake from vFFs report that vFFs make a significant contribution to mean daily intake and the Dietary Reference Intakes (DRIs) of micronutrients [8,9], and they have outlined the risks of excessive intake [10]. Thus, the prevalence of inadequacy observed in Japanese adults may be overestimated, and the prevalence of intakes above the tolerable upper intake level (UL) may be underestimated [3].” (Lines 55-59)

In addition, we added finding of reference [3], as follows; “…, dairy consumers had more adequate calcium intake than non-dairy consumers [3]. However, this study did not consider vFFs, and…” (Lines 41-43)

Methods

  • Did the questionnaire contain any questions related to socioeconomic status?
  • Why were not educational status, income, physical activity, smoking, weight, and BIM measured?

(Response) Thank you for your comment. We understand that socioeconomic status, physical activity, smoking, weight, and BMI affect dietary habits. In this study, we focused to examine the amounts of nutrients consumed from vFF and DS and the usage of them. In addition, it is reported that the number of questions influence reliability of their answers. To reduce the burden on the participants, we prioritized questions and gave up the background information in this study. But we would like to reveal the relationship between vFF and DS use and socioeconomic and other background status in the next study.

  • Were questions related to use in past years? Or different time frame?

(Response) Unfortunately, we did not conduct this survey in the different time frame. There are limited studies on vFF, so we first focused to obtain product information, and check usability of this survey method. We also think that continuous study is important with socioeconomic and dietary intake issues.

  • Please describe sampling method.

(Response) We divided the population into 12 cells by sex and age by 10-year from 20s to 70s and obtained at least 400 responses from each cell. Since the participants were online survey monitors, the responses were collected first-come, first-served basis. We added the following description;

“The population was divided into 12 cells by sex and 10-year age groups, and at least 400 responses from each cell were collected.” (Lines 89-90)

  • Why was not dietary intake assessed by dietary questionnaire? Understanding the intake of various nutrients from different food sources is crucial for evaluating the nutritional status of the Japanese population.
  • It is important to assess the energy intake of participants.

(Response) Thank you for your comment. We agree that it is crucial to evaluate nutritional intake totally; however, researching dietary intake, vFF and DS products, their usages, and background characteristics is a large scale. Before we conduct such large scaled survey, we need to confirm the useful method to obtain maximum information about the use of vFF and DS in this preliminary exploring study. We would like to proceed to the next full-scaled survey.

Results

  • 138: Suggest renaming the section title

(Response) Thank you for your suggestion. We changed the section title to “Nutrient contents of vFF and DS products per daily suggested amount”. (Line 160)

  • 152-154: Rephrase wording is awkward/unclear

(Response) As the reviewer commented, the meaning of the sentence was unclear. We changed the sentence as follows; “The nutrient contents on the label in the reported DS products varied depending on the products, but all vitamins except for vitamin K were contained in almost half of VMS products.” (Lines 174-176)

  • Table 3: why only differentiating based on sex?

(Response) In the preliminary analysis, no significant differences by age were observed, so we only differentiated them based sex. We added the following sentence in the Section 2.4.

“Since no significant differences by age were observed in the preliminary analysis, we differentiated the participants only by sex.” (Lines 142-143)

  • 170: rename subheading

(Response) Thank you for your suggestion. We changed the section title to “Amounts according to the use of vFFs and/or DSs”. (Line 193)

  • 171 – 182: Table 4 is not shown

(Response) We truly apologize for the mistake. We added Table 4 immediately following Table 3.

  • Table 5: Suggest renaming table “based on type of foods” is unclear

(Response) Thank you for your suggestion. We changed the table title to “Usage information for vFFs and DSs according to the use of vFFs and/or DSs”. (Line 238)

  • Significant p-values not displayed in text when describing results

(Response) Thank you for your suggestion. We added p-values.

  • Were there any adverse effects associated with the consumption of fortified foods and, in particular, dietary supplements? Did the participants report any negative outcomes?

(Response) Thank you for your question. Assuming that certain percentages of people experience the adverse effects by using dietary supplements, it is important to collect information about adverse effects. Unfortunately, we did not ask about them this survey, but would like to include the questions asking about negative outcomes next time.

Conclusion

  • 234-236: This sentence is unclear, separate into two parts

(Response) Thank you for your suggestion. We separated the sentence into two sentences as follows;

“This study found that 23.3% of participants used vFFs and/or DSs. This prevalence was higher than what was previously reported, but similar to that in a more recent re-port of Japanese adults [4,6]. However, this prevalence was quite low compared with that of other populations, such as adults in the United States [10], Ireland [14], Finland [15], and adolescents in Poland [9].” (Lines 266-270)

  • 285-287: Discussing nutrient adequacy by age, but this was not evaluated in the paper. Only evaluated nutrient adequacy by sex

(Response) As the reviewer commented, we evaluated nutrient adequacy only by sex not by age. So, we deleted the concerned statements.

  • Since the majority of fortified foods, such as beverages, are considered unhealthy, do you advise individuals to increase their consumption of fortified foods? What are your thoughts on dietary supplements? Is it a good strategy to promote the use of dietary supplements?

(Response) Dietary supplements may be a choice for people who have difficulty in eating adequate amounts of regular foods to supplement nutrients. However, dietary supplements are used by consumers’ own choices with no professionals’ control. In the current situation, where inappropriate use of the dietary supplements concomitantly use with drug by patients and excessive use has been observed, it cannot be easily recommended. On the other hand, food has the advantage that it is difficult to excessively consume. But it is well known that the consumption of snacks and sugar-sweetened drinks contributes significantly to daily energy intake. In addition, it has been suggested that the consumption of nutrient-poor snacks may be associated with unfavorable dietary habits, the association of vFF (nutrient-rich snacks) with these outcomes should be studied.

  • What are the implications of this study?

(Response) To address insufficient nutrient intakes, the use of vFF and DS may be a good alternative. But consumer education to be aware of the use of vFF and/or DS and to use the package labels is needed to be implemented before encouraging the vFF and DS use.

Minor comment

  • vFF abbreviation – is this for voluntary fortified foods? In abstract different definition used (nutrient-fortified foods) then in introduction (voluntary fortified)

(Response) Thank you for your comment. We corrected the statement to voluntary fortified foods in abstract. (Line 14)

Comments on the Quality of English Language

The paper needs English language revision overall as some portions are unclear and there is missing data.

(Response) Thank you for your suggestion. The revised manuscript has undergone English language editing by MDPI.

Again, we thank the reviewers for giving us the opportunity to revise this manuscript and trust that we have been able to do so to their satisfaction.

Best regards,

Chiharu Nishijima

Reviewer 2 Report

Thank you for the opportunity to review the manuscript entitled “Nutrient intake from voluntary fortified foods and dietary supplements in Japanese consumers: Cross-sectional online survey”

This is a study on a topic of great relevance since, as the authors comment, the use of fortified foods and dietary supplements is clearly on the rise. Consequently, the hypothesis put forward about their contribution to the daily intake of vitamins and minerals is interesting. Especially because of the risk of exceeding tolerable upper intake levels. In this respect, users' knowledge of the nutritional information provided on labels is essential. 

However, the manuscript has shortcomings:

- An excessively brief introduction to the topic and a lack of references to key statements (Lines 43-46; 57-58).

- The methodology described indicates several limitations. Some of them are commented on by the authors in the discussion (for example, that described in Lines 78-85), but there are others that are not and raise doubts about how the results have been obtained. This is the case of the description of the questionnaire used (Section 2.2.): the type of questions, the number of questions, whether validation was carried out, etc. are not indicated. When analysing the information on the products (Section 2.3), decisions are made that are not justified by the authors. There is a vague description of the statistical analysis (Section 2.4.).

- As for the results, in several paragraphs it is difficult to understand what the authors want to show, and some inconsistencies have been detected (e.g. there are mistakes when commenting on table 1 - see line 129 which does not correspond to the figures given in the table). In any case, the most remarkable is the use of EAR (estimated average requirement) instead of RDA (recommended daily allowance) as reference values or that table 4 (quoted in line 172) is not found in the manuscript.

Author Response

Thank you for valuable comments and useful suggestion to improve our manuscript. We carefully read them and changed our manuscript by following each comment.

- An excessively brief introduction to the topic and a lack of references to key statements (Lines 43-46; 57-58).

(Response) Thank you for your comment. We added the following descriptions regarding the survey method and finding from the previous studies with references including Lines 43-46.

“In the limited number of studies that reported vFF use in Japan, the prevalence of vFF consumers is reported to be as low as a few percent using the dietary record method [4]; however, approximately 30% was reported by asking about usage habits, regardless of whether the survey was conducted in person, by questionnaire, or online [5-7]. The reasons for these differences are not clear, but the dietary record used in the former study was a one-day survey. The previous studies in other countries have often used the food frequency questionnaire, and assessing habits may be more appropriate to clarify vFF use. In this regard, the prevalence and usage of vFFs in Japan are unclear.” (Lines 47-54)

“Studies on the nutrient intake from vFFs report that vFFs make a significant contribution to mean daily intake and the Dietary Reference Intakes (DRIs) of micronutrients [8,9], and they have outlined the risks of excessive intake [10]. Thus, the prevalence of inadequacy observed in Japanese adults may be overestimated, and the prevalence of intakes above the tolerable upper intake level (UL) may be underestimated [3].” (Lines 55-59)

(Response) For Lines 57-58, we added only references. (Line 72)

- The methodology described indicates several limitations. Some of them are commented on by the authors in the discussion (for example, that described in Lines 78-85), but there are others that are not and raise doubts about how the results have been obtained. This is the case of the description of the questionnaire used (Section 2.2.): the type of questions, the number of questions, whether validation was carried out, etc. are not indicated. When analysing the information on the products (Section 2.3), decisions are made that are not justified by the authors. There is a vague description of the statistical analysis (Section 2.4.).

(Response) Thank you for your comment on the methodology.

Section 2.2

The question used in the screening survey was developed based on the precedingly conducted unpublished survey to ascertain the words in which consumers most correctly identify vFF products. Thus, we added the following sentence to the first paragraph of Section 2.2.

“This question was developed based on the preceding survey to ascertain the words in which consumers most correctly identify vFF products.” (Lines 105-107)

The question to identify the used vFF and DS products was developed in the same way as our previous study for DS products, so we have added representative references. This question has been repeated in 2017-2019 with similar results. However, it has not been validated, so we added the following sentence to the limitation.

“Although the questions about the use and usage of vFFs and DSs were developed while referring the previous studies, they have not been validated.” (Lines 344-345)

Section 2.3

The product analysis was performed similarly to our previous study, so we added representative references. We also added that the analysis was performed by two trained dietitians and the first author confirmed the results.

“This process was conducted by trained dietitians, and the results were confirmed by C.N.” (Lines 124-125)

Section 2.4

We added more descriptions in Section 2.4 as follows;

“Participants were first classified into user, non-user, and not sure groups, and the distributions of characteristics were expressed as percentages; the differences between groups were compared using the chi-squared test according to sex. Regarding the user group, the amounts of vitamins and minerals from vFFs and DSs were presented as the 50th, 75th, 90th, and 97.5th percentiles. Since no significant differences by age were observed in the preliminary analysis, we differentiated the participants only by sex. The usages of vFFs and DSs and the use of the package labels among the user group were compared between “vFF-only”, “DS-only”, and “vFF+DS” subgroups. Differences were examined using the chi-squared test. Statistical analyses were performed using JMP (SAS Institute Inc., Cary, NC, USA) version 13.0 with the statistical significance set to P < 0.05.” (Lines 138-147)

- As for the results, in several paragraphs it is difficult to understand what the authors want to show, and some inconsistencies have been detected (e.g. there are mistakes when commenting on table 1 - see line 129 which does not correspond to the figures given in the table). In any case, the most remarkable is the use of EAR (estimated average requirement) instead of RDA (recommended daily allowance) as reference values or that table 4 (quoted in line 172) is not found in the manuscript.

(Response) Thank you for your comment and we apologize for the mistakes.

Line 129

We agree that the statement in line 129 does not correspond to the results shown in the table. We deleted the sentence “Regarding the dietary type, more participants in the users group were on a specific diet, particularly a high-carbohydrate and high-protein diet.” in the results and the third paragraph of the discussion discussing the diet of vFF and/or DS users.

The use of EAR instead of RDA

In this study, since the data on usual diet consumption was not collected, we presented vitamins and minerals intake from vFF and DS in relation to the percentile of the DRIs instead of using the probability approach to estimate the prevalence of nutrient inadequate intake. However, to estimate the contribution of vFF and DS in the context of complementation, we used the EAR (estimated average requirement) that indicates having a 50% risk of insufficiency.

Table 4 (quoted in line 172)

We truly apologize for the mistake. We added Table 4 immediately following Table 3.

Again, we thank the reviewers for giving us the opportunity to revise this manuscript and trust that we have been able to do so to their satisfaction.

Best regards,

Chiharu Nishijima

Round 2

Reviewer 2 Report

I would like to thank the authors for the modifications/substitutions made that have improved the manuscript.

However, there are still some issues that need to be resolved.

1. Introduction

- Throughout the section, the authors indicate the existence of studies that are not referenced. Therefore, it is essential to add the corresponding citations in lines 39 ([...] "recent studies"), 44 ([...] "in many cases"), 52 ([...] "previous studies") and 67 ([...] "the limited studies").

- Revise the wording related to citation [3] to clarify the suitability of this information in the corresponding paragraph (lines 39-44)

- In the paragraph that has been incorporated in this version in lines 55-57, it should be indicated that these are studies carried out in Europe (or something similar that explains the location of these studies).

2. Materials and Methods

Section 2.1.

- The obtained collection rate (14.1%) should be discussed in the corresponding section ... Is it high/low/similar to other online surveys?

It is important to support the reliability of the conclusions.

- It is suggested that the information on lines 120-121 be incorporated into this section since it is obtained from the company conducting the survey.

- The information for lines 95-101 should be presented in section 2.2, once the questions that allow the distinction to be made between these groups have been described. At this point, the reader does not know what each group "means".

Regarding the 172 participants that the authors have finally classified as "not sure group" (Line 99), please explain why this was done and why they were not excluded from the study.

Section 2.2.

- The authors have incorporated an explanation of how the screening survey was designed (lines 105-107). However, looking at the results, 38.8% of respondents (according to data included in line 97) were misclassified. If this is an exploratory study that they want to continue, as the authors report, is this a misclassification rate that can be accepted? If not, they need to rethink the way in which the screening is carried out, as this question would not be valid. This issue should be discussed in the corresponding section.

- This reviewer has already requested details of the questions included in the full-scale survey. In particular, it would be interesting to have information on:

o   total number of questions included in the questionnaire

o   type of questions: open questions, yes/no questions, participants could choose from a number of options, etc. It is only clear in the case of the question about type of diet (Lines 117-120).

o   estimated time to complete the questionnaire

Section 2.3.

- In lines 132-134, the authors assume as intake the amount suggested by the manufacturer, could you please indicate on what basis you have based this assumption? Is it an assumption that is usually made? It would be sufficient to associate it with some bibliographic citation(s) to justify that this decision is valid. If this is not the case, when the authors indicate it as a limitation, they could say that you should consider including this question (how much do you consume?) in the questionnaire.

3. Results

Section 3.1.

- It is suggested that the authors include a description of the whole sample to provide a picture of the profile of people who have been involved in the study.

- In addition, there are some variables for which statistically significant differences have been obtained that are not addressed in the text.

Section 3.3.1.

- In line 182 the authors indicate "most females", but the figures given in the table do not indicate this.

- This reviewer has already mentioned the difference between the concepts of EAR and RDA/AI. In their reply, the authors indicated that they used the EAR values. However, in this new version, both values are apparently used interchangeably (see column in table 3 and text on line 185). It is important that the authors clarify which value is being used as a reference for evaluating micronutrient intakes from vFFs and DSs and reflect it in the manuscript.

Section 3.3.2.

- For consistency with the title of this paragraph and with tables 5 and 6, replace "type of food" by "use of vFFs and/or DSs" in line 193 and in the title of the table 4.

- There is an mistake in the figure on line 195 (64 instead of 67).

- Given that no corroborating statistical tests have apparently been performed, the authors should review their claims regarding the similarity or otherwise of results between groups (e.g. lines 201-203).

- It is suggested to add a column in Table 4 where UL values are incorporated for better reader tracking.

Section 3.4. y section 3.5.

- It is thought desirable that Tables 5 and 6 include a column reporting the results for the whole group (n=1551) for the different variables.

- It is recommended that the different possibilities for the variables "reasons for use", "factors that affect purchase selection" and "label items that influence purchase selection" be ordered by frequency of response. They will provide the results in a more visual way.

- Authors should consider whether there is important information (related to the statistically significant differences reflected in the tables) that should be incorporated into the text of the sections.

4. Discussion

Along with indications previously indicated that could be interesting to add in this section, it is suggested that authors take into account:

- to mention the possible limitation regarding the representativeness of the sample (there are important socio-demographic data that have not been collected)

- remember that these are intakes based on vFFs and SDs without considering regular food. It implies that the real figures will be higher (with the possible associated risks) and that the possible continuity of the study requires an assessment of the complete dietary intake.

Author Response

Thank you for valuable comments and useful suggestion to improve our manuscript. We carefully read them and changed our manuscript by following each comment.

  1. Introduction

- Throughout the section, the authors indicate the existence of studies that are not referenced. Therefore, it is essential to add the corresponding citations in lines 39 ([...] "recent studies"), 44 ([...] "in many cases"), 52 ([...] "previous studies") and 67 ([...] "the limited studies").

(Response) Thank you for your comment. We added the corresponding citations.

Regarding lines 39 and 44, we modified the sentences to be appropriate; as a result, we added the citations together.

 “Although recent studies on Japanese adults have reported that significant proportions of the population do not meet the recommended levels of intake of some vitamins and minerals, the amounts of vitamins and minerals added to vFF products were not considered in many cases [3-5].” (Lines 39-42)

For lines 52 and 66, we added the citations in lines 53 and 67, respectively.

- Revise the wording related to citation [3] to clarify the suitability of this information in the corresponding paragraph (lines 39-44)

(Response) Thank you for your suggestion. We deleted the wording related to citation [3] since it is suitable to only describe the reports on inadequate intake of vitamins and minerals.

- In the paragraph that has been incorporated in this version in lines 55-57, it should be indicated that these are studies carried out in Europe (or something similar that explains the location of these studies).

(Response) Thank you for your suggestion. We added the location of studies as follows;

“Studies on the nutrient intake from vFFs in Europe report that vFFs make a significant contribution to mean daily intake of Irish adults and the Dietary Reference Intakes (DRIs) of micronutrients in Polish adolescents [11,12], and a study in the United States have reported the risks of excessive intake [13].” (Lines 53-56)

  1. Materials and Methods

Section 2.1.

- The obtained collection rate (14.1%) should be discussed in the corresponding section ... Is it high/low/similar to other online surveys?

It is important to support the reliability of the conclusions.

(Response) Thank you for your comment. The collection rate of online surveys depends on the use of special panels (groups of people with specific criteria, such as a panel of supplement users) that the research company has. However, in this study, we did not use such a panel, but rather used general monitors of the research company, and obtained a similar rate compared to similar surveys. Therefore, we added the following in the corresponding section.

“This collection rate was similar to other Japanese nationwide online survey [14].” (Lines 94-95)

- It is suggested that the information on lines 120-121 be incorporated into this section since it is obtained from the company conducting the survey.

(Response) We agree to the reviewer’s suggestion. We moved the sentence to Section 2.1., Lines 95-96.

- The information for lines 95-101 should be presented in section 2.2, once the questions that allow the distinction to be made between these groups have been described. At this point, the reader does not know what each group "means".

(Response) Thank you for your comment. We moved the information to Section 2.2., Lines 117-125.

Regarding the 172 participants that the authors have finally classified as "not sure group" (Line 99), please explain why this was done and why they were not excluded from the study.

(Response) We did not exclude the participants who were not sure about the use of vFFs because the presence of these people may make it difficult to investigate vFF use; therefore, we tried to understand the difference in characteristics for future study. Thus, we added the purpose to make the “not sure group” as follows;

“In addition, an analysis of product information found that 172 self-identified users did not remember the product information; thus, they were classified as the not sure group to compare the characteristics of those who are aware of the use of vFFs and/or DSs and those who are not.” (Lines 121-124)

Section 2.2.

- The authors have incorporated an explanation of how the screening survey was designed (lines 105-107). However, looking at the results, 38.8% of respondents (according to data included in line 97) were misclassified. If this is an exploratory study that they want to continue, as the authors report, is this a misclassification rate that can be accepted? If not, they need to rethink the way in which the screening is carried out, as this question would not be valid. This issue should be discussed in the corresponding section.

(Response) As the reviewer comments, we agree that this is an important issue to be addressed. We added this issue as limitation as follows;

“Moreover, the question of the screening survey was selected based on the preceding survey; however, the food products that 38.8% of self-reported vFFs and/or DSs users were not nutrient-fortified foods. This is partly because consumers usually choose these foods by reading claims (e.g., high in vitamin C), which cannot tell whether the claimed vitamins and minerals are fortified or naturally occurred until they read the ingredient information. In this regard, we assumed that self-report overestimates the vFFs use. The use of the other method such as food purchase diary [20] may be considered to obtain more accurate information.” (Lines 345-352)

- This reviewer has already requested details of the questions included in the full-scale survey. In particular, it would be interesting to have information on:

(Response) We apologize for not responding to the reviewer's comments.

o   total number of questions included in the questionnaire

(Response) The number of questions were 13 in total: 3 in the screening survey and 10 in the full-scale survey. We added this in Section 2.2. as follows;

“The questionnaire consisted of 3 questions in the screening survey and 10 questions in the full-scale survey.” (Lines 98-99)

In accordance with the number of questions in the screening survey, we added the result of subjective health with the following description of the question in Section 2.2.

“Subjective health was asked because it is reported that dietary supplement use was associated with health status. The health status of response options was “Good”, “Have some health complaints”, and “Have a chronic disease”.” (Lines 111-114)

o   type of questions: open questions, yes/no questions, participants could choose from a number of options, etc. It is only clear in the case of the question about type of diet (Lines 117-120).

(Response) Thank you for your suggestion. We modified the explanation in Section 2.2. as follows;

“For usage information and the use of the package labels, we asked “Number of products in use”, “Frequency of use”, “Duration of use”, “Reasons for use”, “Factors that affect purchase selection”, “Label items that influence purchase selection”, “Understanding of nutrient contents”, and “Calculation of the total amount of nutrients” in a choice question form with response options that shown in the tables.” (Lines 126-130)

o   estimated time to complete the questionnaire

(Response) It took 5 to 13 minutes in our pretest. We added the following sentence in Section 2.2.

“The estimated time to complete the questionnaire was around 10 minutes.” (Lines 99-100)

Section 2.3.

- In lines 132-134, the authors assume as intake the amount suggested by the manufacturer, could you please indicate on what basis you have based this assumption? Is it an assumption that is usually made? It would be sufficient to associate it with some bibliographic citation(s) to justify that this decision is valid. If this is not the case, when the authors indicate it as a limitation, they could say that you should consider including this question (how much do you consume?) in the questionnaire.

(Response) As the reviewer comment, we should have included the question “how much do you consume?”. Although we previously confirmed that 94% of dietary supplement users (for weight loss purpose) took supplements as suggested amount written on the label (Nutrients 2019 11(4):866), we do not know the intake amount of vFFs. When we analyzed the nutrient amount, existing data were not sufficient to estimate the general intake of most food items. Therefore, we decided to use the suggested amounts for all foods. We added this in Section 2.3 and the missing key questions are added in the limitations.

(Materials and Methods) For the quantity of products consumed, we did not ask how much participants consumed and existing data were not sufficient to estimate the general intake of most food items; therefore, we assumed that participants consumed products in the suggested amount written on the label or, for beverages, one bottle. (Lines 141-145)

(Discussion) Fourth, the study did not include a question about the daily amount of consumption. The questionnaire should have included the question “how much do you consume?” as compared to the daily serving size indicated on the package; there may have been participants who consumed less or more vFF and/or DS products than the daily serving size indicated on the packages. (Lines 354-359)

  1. Results

Section 3.1.

- It is suggested that the authors include a description of the whole sample to provide a picture of the profile of people who have been involved in the study.

(Response) Thank you for your comment. Since the description of the target participants in Table 1 was not clear, we corrected as follows; “Table 1 shows the characteristics of 4,933 participants according to the use of vFFs and/or DSs in a full-scale survey.” (Lines 161-162)

Also, we modified the title of Table 1 to “Characteristics of participants in a full-scale survey according to the use of vFFs and/or DSs” (Line 168)

- In addition, there are some variables for which statistically significant differences have been obtained that are not addressed in the text.

(Response) Thank you for your comment. We added “Marital status was significantly different only in males.”. (Lines 164-165)

Section 3.3.1.

- In line 182 the authors indicate "most females", but the figures given in the table do not indicate this.

(Response) Thank you for pointing this out. We meant “more than half of females”, not “most females”. We modified the sentence as follows;

“More than half of the participants consumed vitamins B1, B2, C, niacin, and folic acid from vFFs and/or DSs in both males and females; and also vitamin B12 and iron in females.” (Lines 193-195)

- This reviewer has already mentioned the difference between the concepts of EAR and RDA/AI. In their reply, the authors indicated that they used the EAR values. However, in this new version, both values are apparently used interchangeably (see column in table 3 and text on line 185). It is important that the authors clarify which value is being used as a reference for evaluating micronutrient intakes from vFFs and DSs and reflect it in the manuscript.

(Response) Thank you again for your advice. We deleted all the descriptions about the nutrients specified in AI that are vitamins D, E, K, pantothenic acid, biotin, manganese, chromium, and potassium.

Section 3.3.2.

- For consistency with the title of this paragraph and with tables 5 and 6, replace "type of food" by "use of vFFs and/or DSs" in line 193 and in the title of the table 4.

(Response) Thank you for your suggestion. We corrected as the reviewer suggested. (Lines 204-205 and 230)

- There is an mistake in the figure on line 195 (64 instead of 67).

(Response) Thank you for pointing this out. We corrected to the number to 64. (Line 206)

- Given that no corroborating statistical tests have apparently been performed, the authors should review their claims regarding the similarity or otherwise of results between groups (e.g. lines 201-203).

(Response) Thank you for your comment. We modified the description as follows;

“The amounts of the median intake of nutrients in DS-only users were 3 to 10 times higher than those in vFF-only users for most nutrients.” (Lines 211-213)

- It is suggested to add a column in Table 4 where UL values are incorporated for better reader tracking.

(Response) Thank you for your suggestion. We agree that it is easier to understand if UL values are on Table 4; however, what we intend to show in Table 4 is that the type and amount of nutrients is greater in DSs than in vFFs. Therefore, we decided not to show UL values here.

Section 3.4. y section 3.5.

- It is thought desirable that Tables 5 and 6 include a column reporting the results for the whole group (n=1551) for the different variables.

(Response) Thank you for your suggestion. We added the column “All (n=1151)” in Tables 5 and 6.

- It is recommended that the different possibilities for the variables "reasons for use", "factors that affect purchase selection" and "label items that influence purchase selection" be ordered by frequency of response. They will provide the results in a more visual way.

(Response) Thank you for your suggestion. We rearranged the order of the response in the variables "Reasons for use" in Tables 5 and "Factors that affect purchase selection" and "Label items that influence purchase selection” Table 6.

- Authors should consider whether there is important information (related to the statistically significant differences reflected in the tables) that should be incorporated into the text of the sections.

(Response) Thank you for your comment. We added descriptions about the statistically significant results in Section 3.4 and 3.5 as follows;

(Section 3.4) “The reasons for use were significantly different in most response options.” (Lines 246-247)

(Section 3.5) “Eye-catching words on the front of the package and expiration dates were selected more by vFF-only users (p < 0.001 and p = 0.006, respectively). Nutrition labels, product name, ingredients, and functional claims were selected more by vFF+DS users (p = 0.002, p = 0.015, p = 0.038, and p < 0.001, respectively). Warning labels were selected more by both DS-only and vFF+DS users (p < 0.001). Although 60–70% of individuals, regardless of the use of vFFs and/or DSs, referred to nutrition labels, only 10% of DS-only and vFF+DS users and 4% of vFF users clearly understood the nutrient content.” (Lines 258-265)

  1. Discussion

Along with indications previously indicated that could be interesting to add in this section, it is suggested that authors take into account:

- to mention the possible limitation regarding the representativeness of the sample (there are important socio-demographic data that have not been collected)

(Response) Thank you for your comment. We added description about generalizability in limitation as follows;

“Also, the participants were the online survey monitors and socio-demographic data was not collected; these make generalization difficult.” (Lines 339-340)

- remember that these are intakes based on vFFs and SDs without considering regular food. It implies that the real figures will be higher (with the possible associated risks) and that the possible continuity of the study requires an assessment of the complete dietary intake.

(Response) Thank you for your comment. We agree with the reviewer’s comment. Thus, we added the following sentences;

“In addition, this study focused only on vFF and/or DS consumption without considering regular food, which did not allow the estimation of the contribution of nutrient intake from these products as a proportion of the total intake. This implies that the real figures will be higher (with the possible associated health risks) and that the possible continuity of the study requires an assessment of the complete dietary intake.” (Lines 359-364)

Again, we thank the reviewers for giving us the opportunity to revise this manuscript with valuable comments and suggestion, and trust that we have been able to do so to their satisfaction.

Best regards,

Chiharu Nishijima
